

# Confidence in eating disorder knowledge does not predict actual knowledge in collegiate female athletes

Megan E. Rosa-Caldwell[1,2], Christopher Todden[2], Aaron R. Caldwell[1,2] and Lauren E. Breithaupt[3,4]

[1] Exercise Science Research Center, University of Arkansas at Fayetteville, Fayetteville, AR, United States of America
[2] Department of Behavioral and Health Sciences, Baker University, Baldwin City, KS, United States of America
[3] Department of Psychology, George Mason University, Fairfax, VA, United States of America
[4] Department of Medical Epidemiology and Biostatistics, Karolinska Institute, Stockholm, Sweden

## ABSTRACT

**Background**. Eating disorders are serious psychological disorders with long term health impacts. Athletic populations, tend to have higher incidences of eating disorders compared to the general population. Yet there is little known about athletes' eating disorder knowledge and how it relates to their confidence in their knowledge. Therefore, the purpose of our study was to evaluate collegiate female athletes' eating disorder (ED) knowledge and confidence in their knowledge. 51 participants were recruited from a National Association of Intercollegiate Athletics (NAIA) university in the mid-west and asked to complete a 30-question exam assessing one's knowledge of five different categories related to eating disorders. Confidence in the correctness of answers was assessed with a 5-point Likert-scale (1 = very unconfident, 5 = very confident). A one-way ANOVA was used to determine differences between scores on different categories and overall scores. A simple regression analysis was used to determine if confidence or age was predictive in knowledge scores.
**Results**. The average score of participants was 69.1%, SD = 10.8% with an average confidence of 3.69/5, SD = 0.33. Athletes scored lowest with regards to Identifying Signs and Symptoms of EDs compared to other sub-scores ($p < 0.05$). There was no relationship between knowledge and confidence scores.
**Discussion**. There is limited ED knowledge among collegiate female athletes. This may be problematic as many athletes appear confident in the correctness of their answers despite these low scores. Coaches should be aware of this lack of knowledge and work with clinical practitioners, such as dieticians, team physicians and athletic trainers to educate and monitor their athletes on eating disorders, specifically signs and symptoms.

## INTRODUCTION

Eating Disorders (EDs) are serious mental illnesses affecting millions of individuals worldwide regardless of race, age, nationality, or sex, and are associated with high levels of morbidity and mortality (*Chesney, Goodwin & Fazel, 2014*; *Keshaviah et al., 2014*;

Corresponding author
Megan E. Rosa-Caldwell, mrosa@uark.edu

*Micali et al., 2015*; *Schaumberg et al., in press*). The cumulative lifetime risk of an ED is around 4.6% and an even larger number of individuals (nearly 10%) meet criteria for a subthreshold ED diagnosis (*Hudson et al., 2007*). EDs (including anorexia and bulimia) and subthreshold EDs (not fully meeting diagnostic criteria for either anorexia and bulimia) are more prevalent among female athletes compared to non-athletes, and these disordered behavior can dramatically affect an athlete's health and performance (*Bratland-Sanda & Sundgot-Borgen, 2013*; *Joy, Kussman & Nattiv, 2016*; *Segura-García et al., 2010*; *Sundgot-Borgen & Torstveit, 2010*). For female athletes, this is a particular concern because energy restriction increases the risk for the female athlete triad, referring to three interrelated health threats, consisting of inadequate energy availability, menstrual disorders, and decreased bone mineral density (*De Souza et al., 2014*; *Nattiv et al., 2007*). Considering these pathologies can largely impact future bone and musculoskeletal health as well as quality of life, (*Mueller et al., 2015*; *Palacios et al., 2014*) it is imperative to identify and treat disordered eating behaviors and EDs as early as possible (*Montenegro, 2006*; *Papathomas & Lavallee, 2012*; *Sundgot-Borgen & Torstveit, 2010*).

A substantial amount of research exists on ED risk and prevalence within athletes at large universities (e.g., NCAA Division I and II) (*Abood, Black & Birnbaum, 2004*; *DiPasquale & Petrie, 2013*; *Gutgesell, Moreau & Thompson, 2003*; *Nagel et al., 2000*; *Rosen et al., 1986*; *Sanford-Martens et al., 2005*); (*Sherman et al., 2005*; *Sundgot-Borgen & Torstveit, 2010*; *Turner & Bass, 2001*). However, little research has been completed at institutions with smaller athletic programs, for example National Association of Intercollegiate Athletics (NAIA) universities. NAIA universities (<2,000 students) often lack the resources that are readily available at larger universities, such as access to team-specific physicians, team-specific athletic trainers, or dieticians. Furthermore, recent literature suggests that small universities may have higher prevalence of mental illnesses, including disordered eating (*Eisenberg, Hunt & Speer, 2013*). Therefore, athletes at small universities represent an underrepresented research population with unique requirements for identification and treatment of disordered eating in athletes compared to larger universities.

In general, teammates and coaches spend a significant amount of time together, often times consuming multiple meals a week together. As such teammates and coaches may have the greatest opportunity to identify inappropriate eating behaviors and advocate professional treatment by allied health provider (*Sherman et al., 2005*). However, athletes and coaches generally do not have sufficient knowledge of ED behaviors or appropriate nutrition knowledge (*Govero & Bushman, 2003*; *Torres-McGehee et al., 2011*; *Turk & Prentice, 1999*) . A significant gap in ED knowledge is evident in prior research in coaches (*Torres-McGehee et al., 2012*; *Turk & Prentice, 1999*). This trend of inadequate knowledge is comparable with that of student-athletes (*Torres-McGehee et al., 2011*). These studies suggest that lack of knowledge may be a potential problem for collegiate athletics and amplify the difficulty with identifying problematic eating behaviors before they develop into pathological EDs.

One potential mechanism for lack of knowledge may be due to overconfidence (Dunning Kruger effect). The Dunning Kruger effect, first described in the 1990s (*Kruger & Dunning, 1999*) describes the phenomena where an individual may perceive high confidence in

their knowledge on a subject when in fact, the individual does not have the ability to recognize the limitation of their knowledge. Coaches or teammates who are unable to recognize problematic eating behavior but having confidence in their ability to do can have serious ramifications. However, to our knowledge, no study has specifically investigated how athletes' confidence in their knowledge relates to their actual knowledge in NAIA athletes. Therefore, the purpose of this study was to NAIA female athletes' ED knowledge, confidence in their knowledge, and if confidence positively or negatively predicts ED knowledge. We hypothesized that female athletes would not have sufficient ED knowledge, but despite insufficient knowledge, would have high confidence in their ability to address EDs.

## METHODS

### Participants

All procedures were approved by the Baker University Institutional Review Board. Participants were recruited be email from a small National Association of Intercollegiate Athletics (NAIA) institution in the Midwest. Participants were recruited by email inviting them to participate in the study. Inclusion criteria included: female college students actively participating in University-sponsored athletics. After giving informed consent, athletes were sent a link to the online exam. Exclusion criteria included: non-participation in University-sponsored athletics, not having access to a computer with internet access, or being male. 51 female athletes responded and completed the study.

### Protocols

If an athlete chose to participate she was forwarded the informed consent documentation, which outlined justification for participant recruitment, outlined any risks and benefits, and confirmed confidentiality for all participants. Participants then completed a demographic questionnaire and an exam on eating disorder content knowledge (described below). Athletes were free to take the exam at their convenience; the survey of demographic data and exam took approximately 20 min.

### Instruments

Participants first responded to demographics questions regarding their sport participation and age. Participants then completed a 30 question True/False exam that included questions regarding signs/symptoms, risk factors, etiology, prevention, education, management and treatment of EDs as previously described (*Turk & Prentice, 1999*). The exam has been previously utilized to study the concept of eating disorder knowledge (*Govero & Bushman, 2003*; *Torres-McGehee et al., 2011*; *Turk & Prentice, 1999*). Each subscale is scored individually and total scores are calculated from the sum of correct responses. Additionally, participants were asked to assess their confidence in the correctness of each response (1 = not very confident, 5 = very confident) (*Turk & Prentice, 1999*). Originally formatted by Turk in 1999, the survey has been verified by specialists in the field of athletic training and ED specialist providers (*Turk & Prentice, 1999*). The original survey utilized at 1–4 scale for confidence (*Turk & Prentice, 1999*), however to have more

quantitative data for regression analysis, we changed the scale to 1–5 scale. While the survey has never specifically been validated in previous research, the 30-question true-false exam demonstrated reasonable difficulty ($p = 0.61$, SD = 0.25) despite a somewhat low Cronbach Coefficient $\alpha$ value ($\alpha = 0.57$), suggesting the instrument was a reasonable measure of ED knowledge. Additionally, confidence scores demonstrated good internal reliability with a Cronbach Coefficient $\alpha = 0.80$.

### Statistical analysis

A power analysis was conducted using G*Power to determine an appropriate samples size (*Faul et al., 2009*). While we did not hypothesize a specific $R^2$ for the relationship between confidence and overall score, we *a priori* determined that a $R^2 < 0.15$ was negligible in this population, and therefore used this value to base power calculations. With an effect size of 0.38, a $\beta = 0.80$, and $\alpha = 0.05$, we determined a sample size of approximately 52 was reasonable for this study.

Descriptive statistics were used to describe raw and themed scores among the entire population. Adequate knowledge was determined to be at least 80% correct, has previously described using the same survey (*Govero & Bushman, 2003*; *Torres-McGehee et al., 2011*). Results for each subscale (signs/symptoms, risk factors, etiology, prevention, education, management) were determined by adding the number of correct responses for each question in the subscale, then dividing the total number of questions in that subscale to determine an average percent correct for each subscale. This same process was also used to determine average confidence for all domains.

To determine differences among subscale means and average confidence within subscales, a one-way ANOVA was used with a Tukey-Kramer post-hoc. Additionally, to determine if confidence was predictive of a higher score, simple regression analysis was performed on each of these predictor variables. Upon visual inspection of the data, we noticed older athletes (>20 years old) appeared to score higher than younger athletes (<20 years old); therefore, as an exploratory analysis we also conducted regression analysis on age in relation to overall scores. Cronbach Coefficient $\alpha$ was determined to estimate internal consistency reliability. Significance was denoted at $p < 0.05$. All data were analyzed with Statistical Analysis System (SAS, version 9.3, Cary, NC) and are presented at MEAN $\pm$ STD. Statistical differences are denoted by different letters in figures, where groups that share letters have means that are not statistically different.

## RESULTS

A total of 51 college athletes (mean age = 19.7, SD = 1.3) enrolled in the study and completed all measures. Demographics of sport participation can be found in Table 1. Of the 51 college athletes, the average score of eating disorder knowledge was 69.1%, SD = 10.8% and in general, individuals were fairly confidence in their knowledge (mean = 3.67/5, SD = 0.33). Twelve participants (23.4%) scored an adequate score (>80%), with the remaining participants 39 participants receiving unsatisfactory scores (<80%).

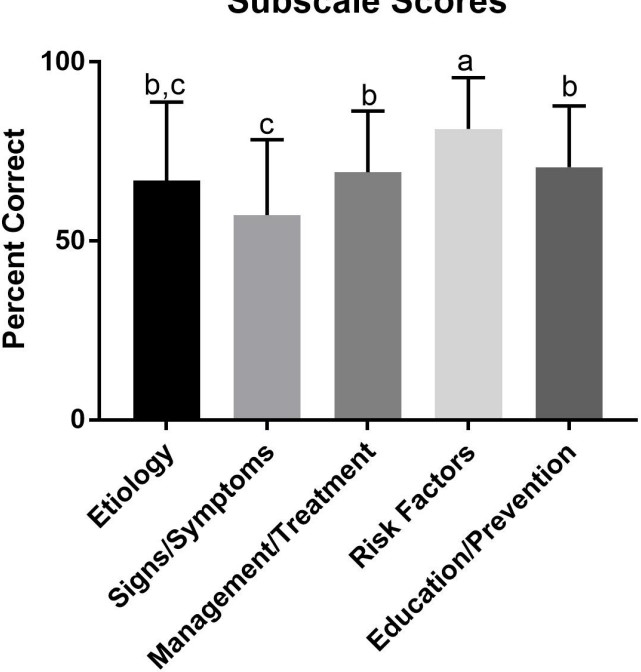

**Figure 1 Scores on domains of eating disorder knowledge.** Subscale scores from the present study. The percent for each subscale was the sum of correct answers divided by the sum of total questions. The percents for all participants were then averaged to determine overall average percent. a, denotes a significant difference from etiology, signs and symptoms, management, and education. b, denotes a significant difference from signs and symptoms and risk factors. c, denotes a significant difference from management, risk factors, and education. The subscales sharing letters are not significantly different ($p > 0.05$).

**Table 1 Descriptive statistics of sports played by athletes in the current study.** Number of individual participants in each sport.

| Sport | n |
|---|---|
| Cheer/dance | 7 |
| Cross country/distance running | 8 |
| Golf | 1 |
| Basketball | 4 |
| Soccer | 9 |
| Tennis | 2 |
| Track-sprints/jumps | 8 |
| Track-throws | 1 |
| Volleyball | 6 |
| Softball | 5 |

Of the five subscales (signs/symptoms, risk factors, etiology, prevention, education, management), identifying risk factors had the highest average percentage (mean = 81%, SD = 14.4%) compared to all other subscales ($p < 0.05$) (Fig. 1), whereas, identifying signs and symptoms subscale was lower than every domain other than the etiology subscale

## Confidence

**Figure 2** **Average confidence of knowledge on various domains of eating disorder information.** Confidence values for subscales from the present study. The confidence for each question was summed and then divided by the total possible confidence (corresponding to a confidence score of 5 for each question), equating to the participant's average confidence for that category. Average confidence was then averaged to determine an overage mean confidence for that category. a, denotes a significant difference from etiology. b, denotes a significant difference from education. Therefore, subscales sharing letters are not significantly different ($p > 0.05$).

(mean $= 57.2\%$, SD $= 21.1\%$, $p < 0.05$). There were no other differences noted between groups (Fig. 1). Interestingly, confidence scores were not different on any subscales (Fig. 2) with the exception of etiology compared to education and prevention.

Regression analysis demonstrated no relationship between confidence and total knowledge scores ($F(1, 49) = 1.94$, $R^2 = 0.038$, $b = 0.063$, $p > 0.05$) (Fig. S1). Through an exploratory analysis we also found a small but statistically significant relationship between participant age and total knowledge scores ($F(1, 49) = 9.17$, $R^2 = 0.16$, $b = .0321$, $p < 0.05$) was noted (Fig. S2).

## DISCUSSION

To our knowledge, we are the first group to analyze eating disorder knowledge specifically in NAIA athletes. The student athletes surveyed lacked eating disorder knowledge; specifically, they scored the lowest on the identification of ED signs and symptoms domain. Further, confidence in perceived knowledge was not predicative of actual knowledge regarding ED knowledge. These results provide preliminary evidence that student athletes do not

recognize their lack of ED knowledge, and therefore, may be ill equipped to recognize unhealthy eating behaviors.

After a preliminary analysis of the data, we noticed a relationship between age and knowledge in our sample; therefore, we conducted a *post hoc* exploratory relationship between age and overall ED knowledge. We found a small relationship between athlete age and increased scores; to our knowledge this is the first study to compare knowledge in relation to athletes' age. It is possible that older students have received knowledge through an outside source during their time at college, such as a nutrition course or a seminar on EDs not specifically sponsored by athletics or some educational program. However, due to limitations of the survey, it is impossible to determine why the older students tended to have higher knowledge compared to younger. However, because this was an exploratory analysis, not the original aims of the study, we urge a cautious interpretation of the data and encourage future research to investigate mechanisms by which age may influence knowledge of EDs.

Overall, many participants did not have appropriate ED knowledge (>80%), however participants still tended to be confident in the accuracy of their knowledge (average confidence > 3.5/5). The results are consistent with previous research utilizing this survey (*Torres-McGehee et al., 2011*; *Turk & Prentice, 1999*), and others that have investigated similar domains (*Torres-McGehee et al., 2012*) demonstrating little knowledge, but high confidence in perceived knowledge. This discrepancy in knowledge and confidence has been seen across many psychological domains (*Pennycook et al., 2017*; *Simons, 2013*). Specifically, for this population, their lack of knowledge in relation to their confidence represents a potential health risk for female student athletes. Depending on the university's resources, there may be limited information available to athletes regarding healthy weight loss and nutrition (*Sherman et al., 2005*). Thus, athletes often ask teammates for nutritional advice (*Bond et al., 2008*; *Reel & Gill, 1996*; *Turocy et al., 2011*). However, the accuracy of this advice may be questionable (*Torres-McGehee et al., 2012*; *Turk & Prentice, 1999*; *Turner & Bass, 2001*; *Wiita & Stombaugh, 1995*), yet our data suggests that they may still be confident in their knowledge, which could contribute to unhealthy eating behaviors.

The ability to identify signs and symptoms was the lowest subdomain in this population. This is concerning as teammates are often in a position to recognize problematic eating behaviors and suggest further treatment due to amount of time spent together (*Sherman et al., 2005*). This may be especially true of athletes at NAIA universities, which may share practitioners with training in nutrition, such athletic trainers, across multiple teams. Athletic trainers monitoring multiple teams may limit opportunities to identify unhealthy eating behaviors. If teammates do not know what behaviors are associated with EDs, then other teammates engaging in pathological eating behaviors may go unnoticed. Moreover, unhealthy eating behaviors may even by female athletes are not familiar with symptoms of EDs and suggests they may not recognize these unhealthy behaviors in teammates. Taken together, a lack in knowledge by fellow teammates as well as limited athletic training staff may lead to unnoticed unhealthy eating behaviors, placing female athletes in NAIA athletic programs at probable risk. However, we should note that this study was only conducted at

one institution, therefore the results are fairly preliminary and this research topic should be further investigated across multiple NAIA universities.

In conclusion, this study reiterates there is poor knowledge of EDs among female collegiate athletes. This is the first study to investigate participants at small collegiate athletic programs, an underrepresented population in the field. Specifically, our data demonstrate that female student athletes clearly lack the knowledge to identify the signs and symptoms of disordered eating in their peers, yet have high confidence in their capacity to do so. Coaches, athletic directors and athletic trainers should be educated on these conditions and educate their respective athletes on EDs. While it should not be the expectation that athletes understand the intricacies of nutrition and EDs, due to the large time spent with teammates, they are likely in the optimal position to notice unhealthy eating behaviors in teammates, and it would be beneficial for them to understand early indicators of problematic eating behaviors and advocate treatment; this may help alleviate the occurrence of full blown EDs or catch disorders early.

### Funding
The authors received no funding for this work.

### Competing Interests
The authors declare there are no competing interests.

### Author Contributions
- Megan E. Rosa-Caldwell conceived and designed the experiments, performed the experiments, analyzed the data, contributed reagents/materials/analysis tools, prepared figures and/or tables, authored or reviewed drafts of the paper, approved the final draft, helped draft the IRB form.
- Christopher Todden conceived and designed the experiments, performed the experiments, contributed reagents/materials/analysis tools, authored or reviewed drafts of the paper, approved the final draft, helped draft the IRB form and submitted final IRB form.
- Aaron R. Caldwell analyzed the data, contributed reagents/materials/analysis tools, prepared figures and/or tables, authored or reviewed drafts of the paper, approved the final draft.
- Lauren E. Breithaupt analyzed the data, authored or reviewed drafts of the paper, approved the final draft.

### Human Ethics
The following information was supplied relating to ethical approvals (i.e., approving body and any reference numbers):

The Baker University Institutional Review board approved this research on Baker University athletes in the 2013–2014 academic year.

## Data Availability

The raw data are provided in the Supplemental Files.

## Supplemental Information

Supplemental information for this article can be found online at http://dx.doi.org/10.7717/peerj.5868#supplemental-information.

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
