# Peer review of "Confidence in eating disorder knowledge does not predict actual knowledge in collegiate female athletes"

_PeerJ, doi:10.7717/peerj.5868_

## Round 0.1 · original submission · Major Revisions

Your manuscript has undergone review and contains several major issues that need resolved before further consideration. If you decide to resubmit your manuscript, please provide a point by point response regarding how each issue raised by each reviewer was addressed in the revised manuscript.

Reviewer 1 ·

Basic reporting

There is an interesting are of research and the study meets ethical requirements.

However, there are some important problems that must be solved before the paper will be found suitable for publication in the scientific journal.

From the Introduction it is not clear what kind of eating disorders are commonly seen in female athletes: how often we can see ED and how often we can see improper nutrition which can significantly effect their health. The Authors should have on mind that not all readers are familiar with this problem in athletes and therefore, it necessary to shortly but precisely introduce it.

Experimental design

There is also not clear if there is an obligatory training for coaches and students on nutrition, nutritional requirements for athletes, relation between nutrition and health, or it is expected that they have some general knowledge as other members of the population should have. Without specific knowledge it can be hard to distinguish between eating habits and eating disorder, which significantly influence the way of treatment. It is an important problem, which significantly influence the questions used in the questionnaire and the answers. The Authors assess the knowledge on etiology, symptoms, treatment etc. It indicated that study participants are expected to possess more that “general knowledge”. It would be important also to present data on subject characteristics related to their university activities (eg. faculty, area of their studies, etc.). The sources of student knowledge on nutrition and eating disorders should be also assessed and presented.

It seems that the questionnaire was used before and how it was validated.

Validity of the findings

It is not clear why the Authors pay attention only o anorexia nervosa, bulimia and starvation, and do not pay attention on eating habits and eating behaviors as well on general knowledge on nutrition and health.

Table 1 should contained also participants characteristics not “simply” numbers of participants involved in different sport activities.

Fig.1 and Fig.2– what “a, b, c “ means???

Fig.3 and Fig.4 – there is no reason to present the figures as no association was found, and the figures do not introduce a specific new information important for the readers

Under Discussion the new data coming form the study should be underlined

Additional comments

There is a lack of refection on the data obtained and what should come out based on the results of this study.

·

Basic reporting

The authors used appropriate English throughout the manuscript. Within the attached review, authors will see additional references to support the manuscript. Sufficient background was provided for most of the aims; however there was additional analysis conducted, but no justification/background to support the analysis (examination of the relationship between age and knowledge). Rationale/background for examining female athletes, needs to consistent throughout the document. Some sections state it may be the responsibility of the athlete to identify signs and symptoms of EDs in other athletes, whereas others it states it should be the dietitian or healthcare provider. With there only being one institution involved in the study, it is unknown what the current healthcare coverage is, so making this assumptions is risky if they indeed have adequate healthcare coverage for student athletes. Does this institution even have access to a dietitian? If not, then what would be the next step? This all needs clarity for such a small sample size, it is almost impossible to generalize to other institutions.

Experimental design

The format of the current methodology section needs organization and clarity, the attached review provides in-depth information on how this section could flow better.

Validity of the findings

Findings are novel however aims need to be clarified and additional background need to be provided for the last aims on examination of the relationship with age and knowledge. If the institution does indeed have adequate medical coverage, this conclusion may be suggestive to focus on prevention and intervention strategies for student athletes for awareness of EDs not necessarily "responsibility" to identify signs and symptoms. This will take the "athletes as the untrained healthcare" provider out of the justification.

Additional comments

NAIA has 250 institutions and 65,000 students athletes; therefore, a sample size of 51 is low. Without a power analysis to determine the effect size and power it is difficult for the reviewer to determine if this is adequate or not. Rationale for the study is conflicting, in the introduction it is clearly stated “instead of team dietitians or athletic training identifying disordered eating behaviors; the responsibility of identifying disordered eating behaviors may fall on the coaches and teammates.” It is recommended the authors be careful with this statement as the institution’s name is in the methods section and may not be reflective of the healthcare providers the student athletes have access too. Due to having only one institution participating in the study, it is important for the authors to be clear on the clinical coverage currently provided for those student athletes or the types of current resources these student athletes have or may not have compared to other institutions in the NAIA. Due to legal issues, administration would never leave the responsibility up to the athletes, so it is recommended authors soften the language for the justification and make sure they have clear evidence prior to making assumptions on the care for the student athletes and who is responsible for the care (needs to be consistent throughout the document).

---

## Round 0.2 · Minor Revisions

We are unable to secure re-review by the second reviewer; however, concerns remain as discussed by the first reviewer.

Reviewer 1 ·

Basic reporting

The manuscript was improved according to previous comments. No additional comment.

Experimental design

The description was improved according to previous comments.
No additional comments.

Validity of the findings

The Results and Discussion was significantly improved, however, it is still necessary to explain in each figure legend what "a, b, c" mean. Maybe it is somewhere at the end of the manuscript but it is hard to find for the reviewer and readers will also have this problem.

It is not necessary to present all data, especially that the study group is small and the results should be called preliminary and additional study in larger group are needed. It is necessary to underline that you found no relationships and figures did not add additional information important for data interpretation.

Additional comments

The manuscript was found to be significantly improved according to previous reviewers' comments, however, still some revision is needed as indicated.
Under Discussion still some reflection is needed as the reviewer is not surprised that older subjects have more knowledge than younger (time is an important factor needed to improve our knowledge in different areas) as well as that "low knowledge" can be accompanied by "quite high" confidence as people who get some knowledge often do not realize the complexity of the problem and the need for cooperation with health professionals.

---

## Round 0.3 · Minor Revisions

The authors still have not fully addressed the issues raised by Reviewer 1. These concerns need to be fully resolved.

Reviewer 1 ·

Basic reporting

The manuscript was improved . No additional comment

Experimental design

The description was improved .
No additional comments.

Validity of the findings

Contrary to the opinion of the authors, the reviewer strongly postulates to clearly indicate under each figure legend differences between which groups are described by letters a, b, c. It is commonly accepted that figure legend contains all information needed to understand the information presented on this figure.

If the authors want to present all data they collected, even data found not significant, such data should be presented in supplementary materials not in the manuscript. Figure 3 should be send to supplementary materials.

Additional comments

The manuscript was found significantly improved. However, the aim of the review is to draw the authors attention to the elements that should be corrected Unfortunately, the authors of this work stubbornly stick to their choices and do not respond to the requests of the reviewer.

Contrary to the opinion of the authors, the reviewer strongly postulates to clearly indicate under each figure legend differences between which groups are described by letters a, b, c.

If the authors want to present all data they collected, even data found not significant, such data should be presented in supplementary materials not in the manuscript. Figure 3 should be send to supplementary materials.

---

## Round 0.4 · accepted · Accept

The manuscript has been sufficiently revised and has been approved for publication by reviewers.

# Reviewer 1 ·

Basic reporting

No additional comments. The manuscript was found suitable for publication.

Experimental design

No additional comments. The manuscript was found suitable for publication

Validity of the findings

No additional comments. The manuscript was found suitable for publication

Additional comments

No additional comments. The manuscript was found suitable for publication